# Development and Multicenter Validation of a Novel Immune-Inflammation-Based Nomogram to Predict Survival in Western Resectable Gastric and Gastroesophageal Junction Adenocarcinoma (GEA): The NOMOGAST

**DOI:** 10.3390/jcm11185439

**Published:** 2022-09-16

**Authors:** Massimiliano Salati, Nicola De Ruvo, Mariano Cesare Giglio, Lorena Sorrentino, Giuseppe Esposito, Sara Fenocchi, Giovanni Cucciarrè, Francesco Serra, Elena Giulia Rossi, Giovanni Vittimberga, Giorgia Radi, Leonardo Solaini, Paolo Morgagni, Giulia Grizzi, Margherita Ratti, Fabio Gelsomino, Andrea Spallanzani, Michele Ghidini, Giorgio Ercolani, Massimo Dominici, Roberta Gelmini

**Affiliations:** 1Division of Oncology, Department of Oncology and Hematology, University Hospital of Modena, 41125 Modena, Italy; 2General, Oncological and Emergency Surgery Unit, University Hospital of Modena, 41125 Modena, Italy; 3Division of Hepato-Bilio-Pancreatic, Minimally Invasive and Robotic Surgery, Federico II University Hospital, 80131 Naples, Italy; 4General Surgery and Advanced Oncological Therapy Unit, AUSL Romagna Forlì Hospital, 47100 Forlì, Italy; 5Division of Oncology, Department of Oncology, ASST di Cremona, Hospital of Cremona, 26100 Cremona, Italy; 6Medical Oncology Unit, Fondazione IRCCS Ca’ Granda Ospedale Maggiore Policlinico, 20122 Milan, Italy

**Keywords:** gastric cancer, prognosis, nomogram

## Abstract

**Background**. More than 50% of operable GEA relapse after curative-intent resection. We aimed at externally validating a nomogram to enable a more accurate estimate of individualized risk in resected GEA. **Methods**. Medical records of a training cohort (TC) and a validation cohort (VC) of patients undergoing radical surgery for c/uT2-T4 and/or node-positive GEA were retrieved, and potentially interesting variables were collected. Cox proportional hazards in univariate and multivariate regressions were used to assess the effects of the prognostic factors on OS. A graphical nomogram was constructed using R software’s package Regression Modeling Strategies (ver. 5.0-1). The performance of the prognostic model was evaluated and validated. **Results**. The TC and VC consisted of 185 and 151 patients. ECOG:PS > 0 (*p* < 0.001), angioinvasion (*p* < 0.001), log (Neutrophil/Lymphocyte ratio) (*p* < 0.001), and nodal status (*p* = 0.016) were independent prognostic values in the TC. They were used for the construction of a nomogram estimating 3- and 5-year OS. The discriminatory ability of the model was evaluated with the c-Harrell index. A 3-tier scoring system was developed through a linear predictor grouped by 25 and 75 percentiles, strengthening the model’s good discrimination (*p* < 0.001). A calibration plot demonstrated a concordance between the predicted and actual survival in the TC and VC. A decision curve analysis was plotted that depicted the nomogram’s clinical utility. **Conclusions**. We externally validated a prognostic nomogram to predict OS in a joint independent cohort of resectable GEA; the NOMOGAST could represent a valuable tool in assisting decision-making. This tool incorporates readily available and inexpensive patient and disease characteristics as well as immune-inflammatory determinants. It is accurate, generalizable, and clinically effectivex.

## 1. Introduction

Gastric and gastroesophageal junction adenocarcinoma (GEA) remains responsible for a considerable health burden worldwide, with 1,089,103 new cases and 768,793 deaths in 2020, ranking fifth for incidence and fourth for mortality globally [1]. Despite a steady decline in new cases and deaths from GEA since 1930, an increase in incidence among individuals aged <50 years has been reported in recent decades [2,3]. While in Eastern countries, the implementation of endoscopic screening has resulted in earlier diagnosis and better outcomes, in the West, most GEAs are diagnosed at an advanced stage with fewer chances of a cure.

The treatment paradigm for a third of Western patients presenting with potentially resectable disease relies on a multimodality strategy, including perioperative FLOT chemotherapy as the most commonly adopted and guideline-endorsed choice [4,5,6]. Nonetheless, more than half of patients still experience relapse and succumb to their disease. In contrast, others who are cured by surgery alone can be overtreated by adding pre- and postoperative chemotherapy [7].

Currently, with the lack of established biomarkers, both prognostication and treatment decisions are driven by patient-related factors and clinicopathologic staging in clinical practice. However, the TNM system presents several drawbacks that limit its accuracy, such as the inability to capture interpatient heterogeneity. Patients falling within the same class may have significantly different outcomes, including a finite number of stages and excluding biological determinants [8].

Recently, two gene signatures showed promising results in stratifying patients based on prognosis and treatment response, both in Western and Eastern GEA [9,10]. More interestingly, the robust positive prognostic effect and the potential predictive value of the MSI-h status is being prospectively explored in ongoing trials of resectable GEA [11,12]. In recent years, prognostic tools such as nomograms have gained popularity among clinicians thanks to their user-friendly interfaces, easy accessibility/availability, and performance against conventional predictive tools. Notably, nomograms, pictorial representations of a statistical predictive model generating a numerical probability of a clinical event, fulfill the demand for an integrated model by incorporating both clinical and biological variables and patient and disease characteristics [13].

In the present study, we aimed to construct and validate a novel prognostic nomogram (NOMOGAST), enabling a more accurate estimate of individualized risk to inform clinical decision-making in resectable GEA.

## 2. Materials and Methods

Electronic medical records of patients undergoing curative-intent gastrectomy for GEA between 2010 and 2018 at three Cancer Centers (Modena, Forlì, Cremona) were retrospectively reviewed.

Inclusion criteria were: histologically confirmed diagnosis of gastric or gastroesophageal junction (Siewert type II and III) adenocarcinoma, locally advanced resectable disease (c/uT3-T4 and/or N-positive), no clinical evidence of distant metastases according to the 8th Edition of the International Union against Cancer tumor–node–metastasis classification, and upfront curative-intent surgical resection followed by adjuvant chemotherapy. Locally advanced unresectable tumors that became operable following neoadjuvant treatment and those treated with radiotherapy (either preoperatively or postoperatively) were excluded.

Potential prognostic parameters were retrieved before surgery based on prior research evidence and/or sound clinical reasoning. As such, the following variables were collected for further analysis: age, gender, ECOG PS, BMI, tumor site, disease stage, T status, N status, angioinvasion, administration of chemotherapy (both perioperative and adjuvant), hematological and biochemical parameters, including white blood cell count (cell/µL), neutrophil count (cell/µL), lymphocyte count (cell/µL), hemoglobin (gr/dL), platelet count (cell/µL), bilirubin (mg/dL), alkaline phosphatase (ALP; IU/L), lactate dehydrogenase (LDH U/L), alanine aminotransferase (ALT; IU/L), aspartate aminotransferase (AST; IU/L), albumin (g/dL), carbohydrate antigen 19-9 (CA 19-9) (U/mL), and carcinoembryonic antigen (CEA) (ng/mL). The prognostic nutritional index (PNI) was calculated as follows: 10 × serum albumin concentration (g/dL) + 0.005 × peripheral lymphocyte count (number/mm^2^), while the systemic inflammatory index (SII): neutrophil count x platelet count/lymphocyte count. The study protocol conformed to the ethical guidelines of the 1975 Declaration of Helsinki. Data were collected under protocol 1186-2018/OSS/AOUMO, reviewed and approved by the Area Vasta Emilia Nord Ethics committee.

### Statistical Analysis, Model Development and Validation

This study aimed to develop a predictive model of overall survival (OS) in patients undergoing curative-intent surgery for gastric and gastroesophageal junction adenocarcinoma. For inferential purposes, a Cox proportional hazards regression analysis was performed on the whole dataset to identify prognostic factors of OS. Variables presenting at univariable analysis a *p*-value < 0.1 were entered into the multivariable analysis. The effects were expressed as a hazard ratio (HR) with a 95% confidence interval (95% CI).

Using the variables selected above, the predictive model was developed in a training cohort composed of patients from the Modena and Forli Cancer Centers and then externally validated in a cohort of patients from the Cremona Cancer Center (validation cohort). Validation was assessed through the analysis of the model’s discrimination ability and calibration. Only complete cases for the variables of interest were used for model development and validation. Missing data were not imputed. Model discrimination was evaluated using the Harrell c-index. The value of the Harrell c-index fluctuates between 0.5 and 1.0, with 0.5 representing random chance and 1.0 representing a totally corrected discrimination. Model calibration was analyzed by inspecting calibration curves to determine whether the predicted and actual survival were in concordance and using the integrated calibration index. Calibration was evaluated for predictions of OS at 3 and 5 years. The clinical usefulness of the derived nomogram was assessed through a decision curve analysis [14,15].

The model was graphically represented as a nomogram to ease its application, using the package rms RMS (Regression Modeling Strategies, ver. 5.0-1) in R software, Nashville TN, USA. Continuous variables are presented as a mean (standard deviation) or median (interquartile ranges) and are compared using the Wilcoxon Mann-Whitney test or the Student’s t-test according to the data distribution. Categorical variables are expressed as frequency and associated percentages and compared using the Pearson Chi-square test and Fisher’s exact test. The Kaplan–Meier method was applied to assess survivor functions. All statistical analyses were performed by the statistical R software version 3.3.2 (http://www.r-project.org/ accessed on 21 June 2022). All *p*-values were two-sided, and the statistical significance was *p* < 0.05.

## 3. Results

### 3.1. Patient Population and Prognostic Factors

Overall, medical charts of 641 patients with radically resected GEA from three referral Cancer Centers were retrospectively reviewed. Among them, 336 patients completely fulfilled the inclusion criteria and were thus included in the analysis. Specifically, 185 patients from the Modena and Forlì Cancer Centers and 151 from the Cremona Cancer Center made up the TC and VC, respectively. The median age at diagnosis was 70.7 and 73.8 years in the former vs. latter group, with more than 50% of patients displaying symptomatic disease and impaired general health conditions at presentation in both cohorts (ECOG PS ≥ 1; *p* = 0.46). A slightly higher proportion of node-positive disease were recorded in the VC (74.8% vs. 63.8%), while GEA from the TC more commonly presented angioinvasion (45.4% vs. 33.1%). Stomach antrum and corpus were the most frequently involved gastric subsites, accounting for more than two-thirds of cases.

Other baselines demographic and clinicopathologic features by study cohort are summarized in Table 1. The baseline cumulative hazards of the training and validation cohorts appear to overlap (Figure 1A,B).

When investigating potential prognostic factors in the two cohorts, the following covariates were included in the univariate analysis for OS: age, gender, BMI, ECOG PS, gastric subsite, CEA, CA19.9, T, N, log NLR (neutrophil-to-lymphocyte ratio), and vascular invasion. Among them, ECOG PS ≥ 1 (HR, 1.97, 95% CI, 1.24 to 3.11, *p* = 0.004), vascular invasion (HR, 2.25, 95% CI, 1.41 to 3.60, *p* = 0.001), log (Neutrophil/Lymphocyte ratio) (1.84, 95% CI, 1.30 to 2.60, *p* = 0.001), and nodal status (HR, 1.99, 95% CI, 1.05 to 3.76, *p* = 0.034) remained independent predictors for OS at the multivariate analysis (Table 2).

### 3.2. Prognostic Model Development and Validation

A model based on these independent prognostic factors was developed, and a nomogram was constructed to estimate 3-year and 5-year OS. The prognostic model presented a fair discriminative ability, with c-Harrell index values of 0.71 and 0.76 in the training and validation cohorts, respectively (Figure 1A,B). The discriminative model ability was also corroborated by the observed survival of patients in the validation cohort when grouped by 25 and 75 percentiles of the probability of survival (Log-rank test, *p* < 0.001, Figure 2). The model also presented an optimal calibration, with plots demonstrating a concordance between the predicted and actual survival in the validation cohorts (Figure 3).

A decision curve analysis was plotted, depicting the clinical utility associated with the clinical application of the model at three years, expressed in terms of net benefit (Figure 4). 

The nomogram is graphically represented in Figure 5, where each variable is listed separately, with a corresponding number of points assigned to a particular magnitude of the variable. Then, the cumulative point score for all the variables is matched to a scale of the outcome of 3- and 5-year OS.

Moreover, the NOMOGAST displayed a superior net benefit compared to the TNM staging system, which is the gold standard for prognostication in oncology, for all the threshold probabilities (Figure 6).

## 4. Discussion

In this study, we developed and externally validated a novel nomogram named NOMOGAST to estimate 3- and 5-year individual OS in Western patients with resectable GEA. Interestingly, this prognostic tool incorporates readily available and inexpensive variables reflecting patients’ characteristics (ECOG PS), biological tumor aggressiveness (nodal status and angioinvasion), and host immune-inflammation status (NLR).

In resectable GEA, both the outcome prediction and treatment decision are dictated by well-established clinicopathologic features. Accordingly, international guidelines recommend multimodality treatment strategies for stage IB-III GEAs [4]. However, a marked heterogeneity exists concerning the risk of relapse and death for patients in the same TNM subgroup. Moreover, TNM staging only generates a finite number of prognostic classes. This staging system does not account for biological determinants with a known impact on survival, such as histological and molecular subtype, grading, and lymphovascular invasion.

On the other hand, prognostic multigene signatures have been suggested to be helpful tools for the risk stratification of GEA patients treated with a multimodality strategy. A seven-gene signature (i.e., CDH1, ELOVL5, EGFR, PIP5K1B, FGF1, CD44v8, and TBCEL) was developed in a Western cohort of 84 patients treated with neoadjuvant chemotherapy that identified a high- and low-risk group with a median OS of 10.2 and 80.9 months (*p* < 0.0001), respectively [10]. Additionally, a four-gene real-time RT-PCR assay, including GZMB, WARS, SFRP4, and CDX1, has been shown to differentiate between three groups with different survivals (low, intermediate, and high risk) [9]. Additionally, it predicts the response to adjuvant chemotherapy (benefit and no benefit) in stage II-III resected Asian GEA after D2 gastrectomy. Although promising, these molecular tools are challenging to implement in daily clinical settings because of high costs, limited accessibility, and the lack of a significant prospective validation. Moreover, their applicability to different ethnic populations remains to be assessed.

In recent years, nomograms have been proposed as clinical tools for assisting in prognosis estimation and treatment decisions, with the potential to be more informative and accurate than conventional TNM/AJCC staging. In GEA, several prognostic nomograms were published that were developed on presurgical or pathologic variables, within heterogeneous patient populations and without accounting for biological determinants. Moreover, some of them incorporate lots of covariates, thus potentially jeopardizing the practicality of these devices [16,17].

Following a rigorous methodological approach applied to a broad spectrum of biologically relevant determinants, our study built a comprehensive prediction tool incorporating patient- and cancer-specific characteristics and factors reflecting immune inflammation. Regarding patient-related factors, our findings align with a large body of literature highlighting a robust and consistent prognostic significance for ECOG PS across a broad spectrum of cancer types, including GEA. Concerning disease-related features, in addition to the well-established negative value of vascular invasion, we confirmed in a large real-world cohort the poor prognostic significance of lymph node metastasis, which has been found to be the only independent predictor of survival following neoadjuvant chemotherapy plus resection in the MAGIC trial [8]. More interestingly, our model also included NLR, based on its independent prognostic significance. This biological marker, an increasingly recognized prognostic factor in oncology, reflects a protumorigenic inflammatory switch and an impaired response by the host immune system [18,19].

We believe that the strengths of our prognostic model are severalfold. Firstly, it is a 4-tier tool incorporating well-established clinicopathologic factors but also immune-inflammatory markers. Secondly, the model performs well and has been successfully validated in a large and independent cohort with similar features. Thirdly, the variables included in the model are readily available and inexpensive, thus fostering its applicability in the clinic. Of note, the NOMOGAST performed better than the TNM staging system in predicting the prognosis of GEA.

At the same time, we acknowledge that the study’s retrospective nature precluded us from evaluating emerging biomarkers such as MSI status. Then, excluding patients receiving neoadjuvant treatment makes this model not directly generalizable to those treated perioperatively. However, it allowed us to avoid treatment’s impact on parameters such as neutrophils and lymphocytes.

In conclusion, by estimating the risk of death in a presurgical setting, the NOMOGAST could assist clinicians in discussing with patients the prognosis and risk-to-benefit ratio of surgery and adjunctive treatment modalities. Therefore, it represents an accurate (well calibrated with a good discriminative ability) and clinically effective tool for implementation in the clinic in order to inform decision-making and improve prognostic accuracy for patients with resectable GEA.

## Figures and Tables

**Figure 1 jcm-11-05439-f001:**
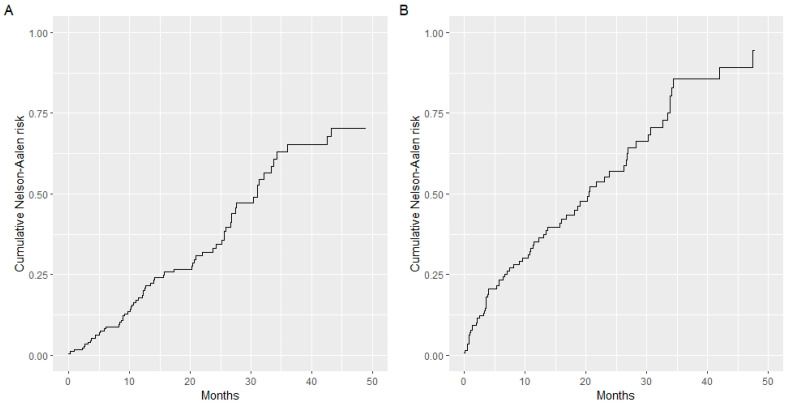
Baseline cumulative hazards of the training and validation cohorts. c-Harrell index values: 0.71 training cohort (**A**); 0.76 validation cohort (**B**).

**Figure 2 jcm-11-05439-f002:**
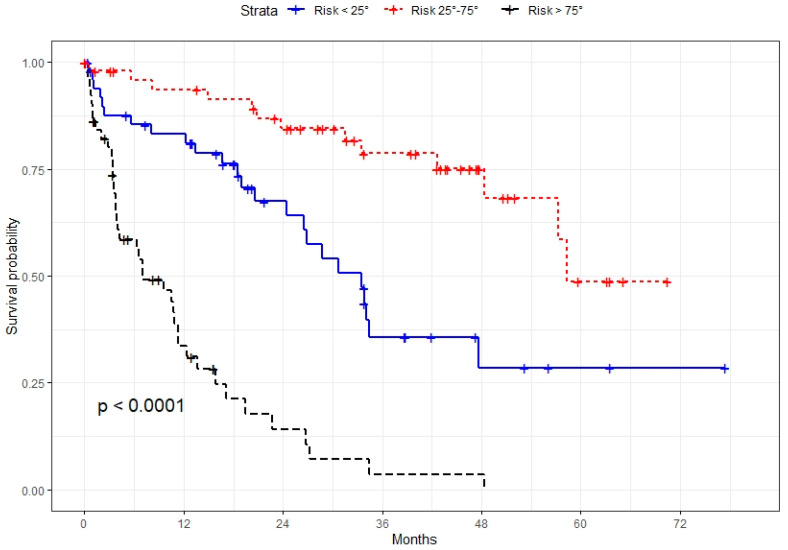
Observed survival of patients in the validation cohort grouped by 25 and 75 percentiles of the probability of survival (Log-rank test, *p* < 0.001).

**Figure 3 jcm-11-05439-f003:**
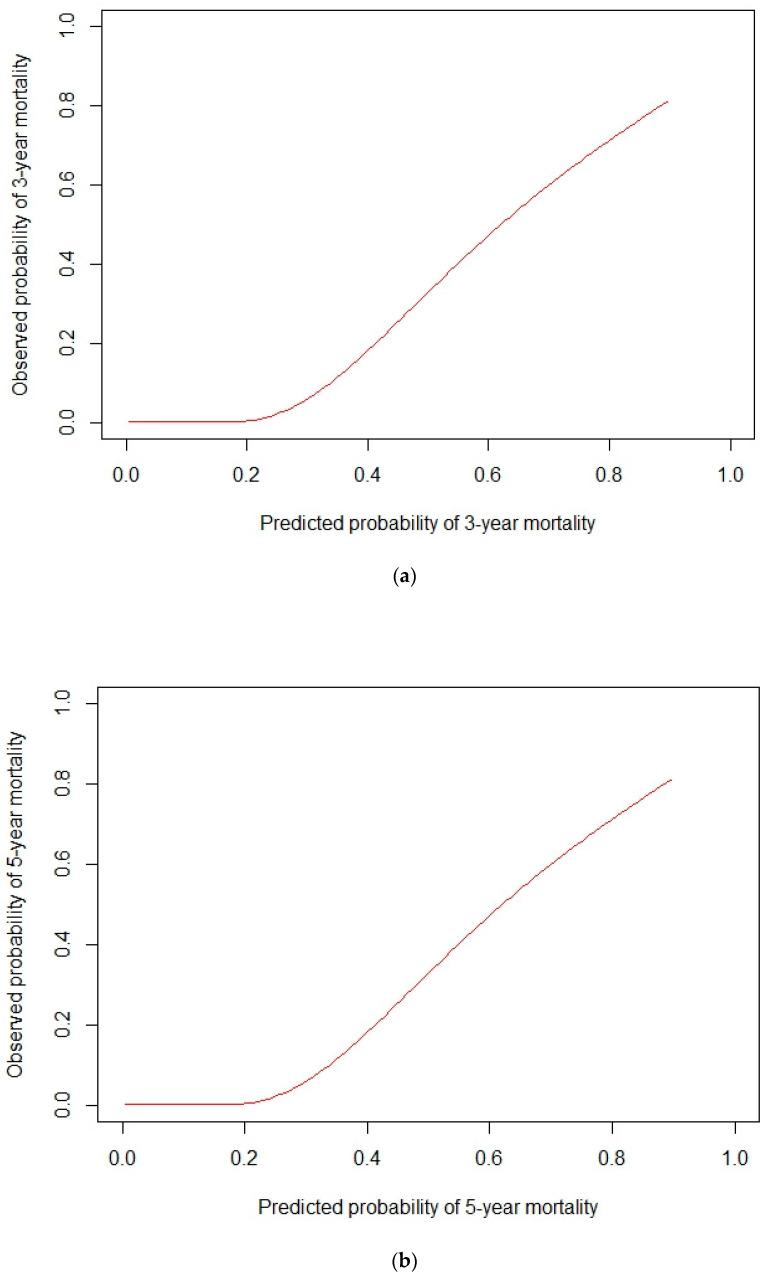
Calibration curves of the prognostic model. Plot demonstrates concordance between the predicted and observed survival in the validation cohorts at (**a**) 3 and (**b**) 5 years.

**Figure 4 jcm-11-05439-f004:**
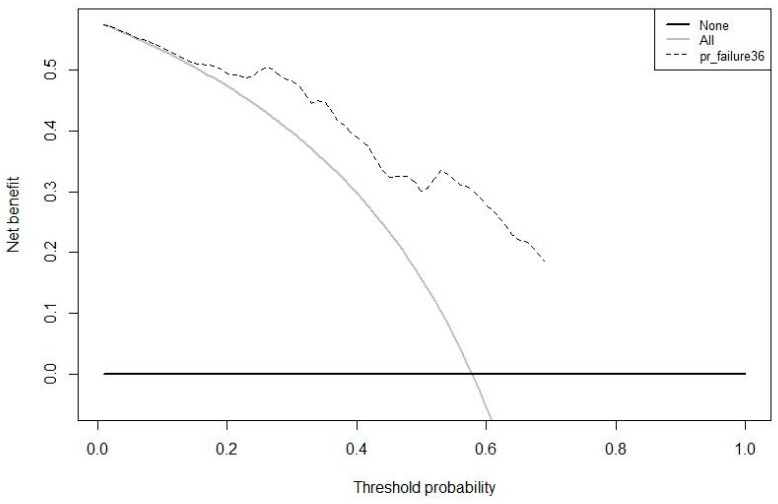
Decision curve analysis. Clinical utility associated with the clinical application of the model at three years, expressed in terms of net benefit.

**Figure 5 jcm-11-05439-f005:**
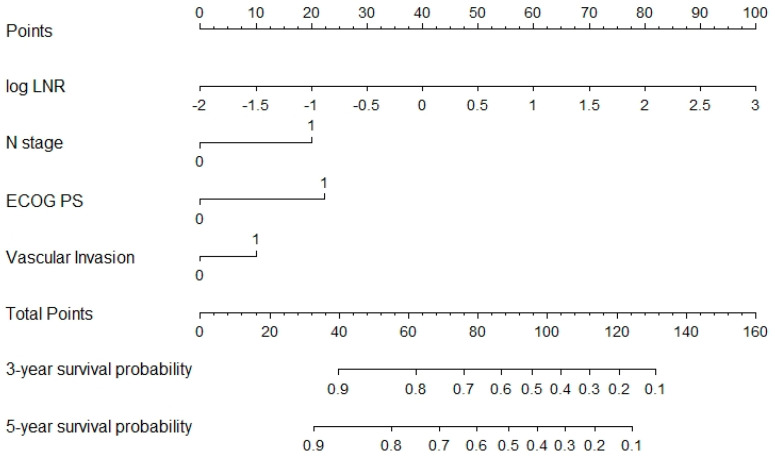
NOMOGAST. A novel nomogram is graphically represented; each variable is listed separately, with corresponding number points assigned to a particular magnitude of the variable. The cumulative point score for all the variables is matched to a scale of the outcome of 3- and 5- year OS.

**Figure 6 jcm-11-05439-f006:**
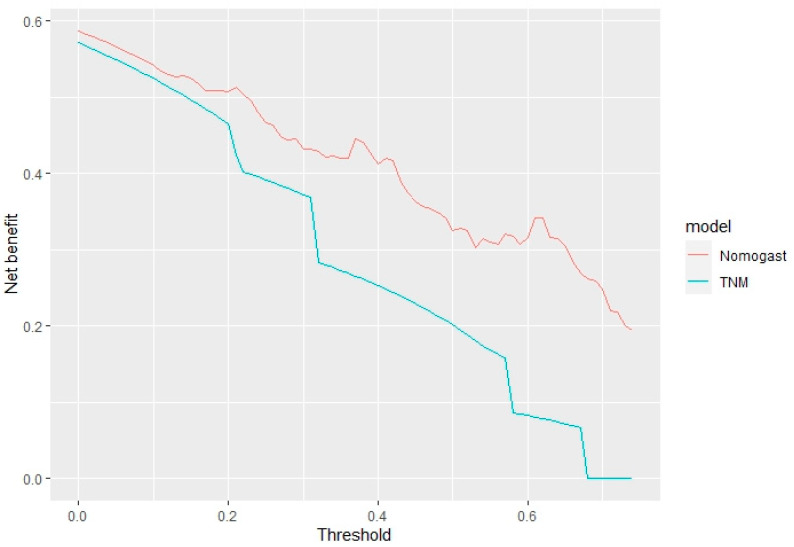
Comparison of clinical utility between the NOMOGAST and TNM staging systems.

**Table 1 jcm-11-05439-t001:** Training and validation cohorts.

	Training (*n* = 185)	Validation (*n* = 151)	*p* Value
**Age** Mean (SD)	70.7 (11.3)	73.8 (9.7)	0.009
**Gender**			0.188
Female	83 (44.9%)	57 (37.7%)	
Male	102 (55.1%)	94 (62.3%)	
**ECOG PS**			0.466
0	87 (47.0%)	65 (43.0%)	
≥1	98 (53.0%)	86 (57.0%)	
**Sites**			0.008
Anastomosis	1 (0.5%)	1 (0.7%)	
Angulus	0 (0.0%)	8 (5.3%)	
Antrum	65 (35.1%)	53 (35.3%)	
Cardias	25 (13.5%)	18 (12.0%)	
Body	78 (42.2%)	56 (37.3%)	
Fundus	12 (6.5%)	5 (3.3%)	
Linitis	2 (1.1%)	0 (0.0%)	
Stump	1 (0.5%)	1 (0.7%)	
Pylorus	1 (0.5%)	8 (5.3%)	
**CEA** Mean (SD)	5.6 (19.3)	7.1 (20.1)	0.191
**CA 19-9** Mean (SD)	46.6 (182.4)	127.8 (784.6)	0.270
**T stage**			0.825
0	0 (0.0%)	0 (0.0%)	
1	33 (17.8%)	18 (11.9%)	
2	40 (21.6%)	20 (13.2%)	
3	64 (34.6%)	61 (40.4%)	
4	44 (23.8%)	50 (33.1%)	
IS	4 (2.2%)	2 (1.3%)	
**N stage**			0.030
0	67 (36.2%)	38 (25.2%)	
1	118 (63.8%)	113 (74.8%)	
**log NLR**			0.044
Mean (SD)	0.9 (0.6)	1.1 (0.8)	
**Vascular Invasion**			0.022
0	101 (54.6%)	101 (66.9%)	
1	84 (45.4%)	50 (33.1%)	

**Table 2 jcm-11-05439-t002:** Univariate and multivariate Cox regression analysis of factors related to OS.

Variable		All	HR (Univariable)	HR (Multivariable)
Age	Mean (SD)	72.2 (10.6)	1.03 (1.01–1.04, *p* < 0.001)	1.01 (0.99–1.04, *p* = 0.163)
Gender	Female	196 (42.5)	-	-
	Male	265 (57.5)	0.97 (0.74–1.28, *p* = 0.847)	-
BMI	Mean (SD)	25.6 (5.0)	1.00 (0.96–1.04, *p* = 0.978)	-
ECOG PS	0	195 (43.5)	-	-
	≥1	253 (56.5)	2.21 (1.63–3.01, *p* < 0.001)	1.97 (1.24–3.11, *p* = 0.004)
Sites	anastomosis	3 (0.7)	-	-
	angulus	16 (3.5)	0.50 (0.21–1.23, *p* = 0.133)	-
	Antrum	163 (35.4)	0.80 (0.60–1.07, *p* = 0.134)	-
	Cardias	55 (12.0)	1.21 (0.82–1.78, *p* = 0.339)	-
	Body	180 (39.1)	0.76 (0.57–1.01, *p* = 0.057)	-
	Fundus	22 (4.8)	0.66 (0.35–1.25, *p* = 0.203)	-
	Linitis	2 (0.4)	0.00 (0.00–Inf, *p* = 0.993)	-
	Stump	6 (1.3)	0.00 (0.00–Inf, *p* = 0.989)	-
	Multifocal	1 (0.2)	34.57 (4.42–270.60, *p* = 0.001)	-
	Pylorus	12 (2.6)	0.97 (0.43–2.19, *p* = 0.944)	-
CEA	Mean (SD)	7.1 (25.7)	1.01 (1.00–1.01, *p* = 0.055)	1.00 (0.97–1.02, *p* = 0.783)
CA 19-9	Mean (SD)	72.1 (485.5)	1.00 (1.00–1.00, *p* = 0.018)	1.00 (1.00–1.00, *p* = 0.288)
T stage	IS	8 (1.7)	NA (NA–NA, *p* = NA)	-
	1	81 (17.6)	2.69 (0.36–20.04, *p* = 0.335)	-
	2	108 (23.4)	4.61 (0.64–33.45, *p* = 0.130)	-
	3	158 (34.3)	6.81 (0.95–49.09, *p* = 0.057)	-
	4	106 (23.0)	8.18 (1.13–59.18, *p* = 0.037)	-
N stage	0	163 (35.7)	-	-
	≥1	293 (64.3)	1.93 (1.39–2.66, *p* < 0.001)	1.99 (1.05–3.76, *p* = 0.034)
log NLR	Mean (SD)	1.0 (0.7)	1.92 (1.55–2.37, *p* < 0.001)	1.84 (1.30–2.60, *p* = 0.001)
Vascular Invasion	Absent	231 (60.5)	-	-
	Present	151 (39.5)	3.04 (2.21–4.18, *p* < 0.001)	2.25 (1.41–3.60, *p* = 0.001)

## Data Availability

The data presented in this study are available on request from the corresponding author. The data are not publicly available due to the Italian legislation on the right to privacy.

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
