# Peer review of "Development and Multicenter Validation of a Novel Immune-Inflammation-Based Nomogram to Predict Survival in Western Resectable Gastric and Gastroesophageal Junction Adenocarcinoma (GEA): The NOMOGAST"

_jcm, 2022, doi:10.3390/jcm11185439_

Round 1

Reviewer 1 Report

In this paper, the author retrospectively collected the data of gastric cancer patients in three centers, searched for independent factors related to prognosis through survival analysis, and constructed a nomogram model to further predict the prognosis of gastric cancer patients. However, at present, a large number of articles have carried out the similar analysis. This draft does not show strong innovation, and there are some problems that need further revision:

1. The research object mentioned in the title of this manuscript is resectable gastroesophageal adenocarcinoma (GEA), but the data of GEA and other gastric cancers are collected in the text. The author should clarify the research object to avoid misunderstanding.

2. In the results of this study, T stage was not included in nomogram model because it was not an independent prognostic factor for patients with gastric cancer. However, as a recognized factor affecting the prognosis of gastric cancer, T-stage should be further incorporated into the model.

3.In this study, nomogram model is constructed and verified, but the verification curve is not provided. It is suggested to supplement

4. The authors of this study did not show the effectiveness of this model in predicting the prognosis of patients with gastric cancer. It is suggested to compare the predictive effectiveness to the existing models or TNM staging

5. This study adopts the 7th edition of UICC's TNM staging of gastric cancer. However, 8th edition should be considered. 

Author Response

In this paper, the author retrospectively collected the data of gastric cancer patients in three centers, searched for independent factors related to prognosis through survival analysis, and constructed a nomogram model to further predict the prognosis of gastric cancer patients. However, at present, a large number of articles have carried out the similar analysis. This draft does not show strong innovation, and there are some problems that need further revision:

  1. The research object mentioned in the title of this manuscript is resectable gastroesophageal adenocarcinoma (GEA), but the data of GEA and other gastric cancers are collected in the text. The author should clarify the research object to avoid misunderstanding.

This research article is focused on gastric and gastroesophageal junction adenocarcinoma (Siewert II and III), which are collectively named as gastroesophageal adenocarcinomas (GEAs). The title and the “Materials and Methods” section have been modified accordingly.

  1. In the results of this study, T stage was not included in nomogram model because it was not an independent prognostic factor for patients with gastric cancer. However, as a recognized factor affecting the prognosis of gastric cancer, T-stage should be further incorporated into the model.

Thank you for your observation. Following your suggestion, we developed again the predictive model by adding, as requested, the T status as predictor. In the figure below, the net benefit of this “extended model” (green line) is compared with the “original” Nomogast predictive model.   As it can be seen, adding the t status does not clearly improve the model in terms of Net Benefit.

3.In this study, nomogram model is constructed and verified, but the verification curve is not provided. It is suggested to supplement

Thank you for your comment. The nomogram is not more than a graphical representation of the prognostic model developed in the paper. For this prognostic model, we have provided a calibration curve (fig. 3) and a graph demonstrating the ability of the model to stratify patient survival (Fig. 2). The c-Harrell has been provided in the text as measure of model discriminative ability. In addition to these classical methods of model validation, we have analysed the clinical impact of the model using decision curve analysis, thus showing the net benefit associated with the use of the model.

  1. The authors of this study did not show the effectiveness of this model in predicting the prognosis of patients with gastric cancer. It is suggested to compare the predictive effectiveness to the existing models or TNM staging

Thank you. In the figure above we compare the net benefit of the nomogast predictive model with a model based only on the TNM staging. As it can be seen, the Nomogast is associated with a net benefit which is superior to that of TNM for all threshold probabilities.

  1. This study adopts the 7th edition of UICC's TNM staging of gastric cancer. However, 8th edition should be considered.

This has been modified in the text.

Figure 5. Comparison of clinical utility between the NOMOGAST and the TNM staging system.

Please find Figure 5 attached.

Reviewer 2 Report

Article "Development and multicenter validation of a novel immune inflammation-based nomogram to predict survival in Western resectable gastroesophageal adenocarcinoma (GEA): the NOMOGAST” by Salati et al. is an interesting work on an important subject i.e., the GEA relapses (>50%) after curative-intent resection. The authors directed their scientific efforts to validate a nomogram for more accurate individualized risk estimation in resected GEA, a topic that is not yet elucidated and remains open.

Based on the analysis of the medical records of a training cohort (TC) and a validation cohort (VC) of patients undergoing radical surgery for c/uT2-T4 and/or node-positive GEA, the authors have collected the variables considered potentially interesting. Effects of prognostic factors were evaluated, and a nomogram was drawn using the R software's package Regression Modeling Strategies (ver. 5.0-1), and the performance of the prognostic model was evaluated and validated by Salati et al.

Initial data of the two cohorts as well as the results of the modelling are presented and discussed in two tables and 4 figures. The final conclusions are presented. The article is not loaded with unnecessary information.

However, as minor problems, I would suggest the following corrections:

1.       To more explicitly introduce any abbreviations used right from the start, and a final table of Abbreviations is very welcome.

2.       References should be presented throughout the article more accurately, enclosed in brackets and followed by a full stop.

So YES, that's right    .....     [1].

                         .........         [2-3].

Please, correct lines: 77, 79, 85, 88, 94, 96, 98 and so on …, until the end of the article!!!

3.       Discussions should be improved by showing all model shortcomings, how results were compared, and the limitations imposed by statistics and the model.

4.       Final conclusions should be better punctuated, to present the real scientific value of this model more accurately.

5.       Overall, the study is interesting and deserves to be published with these corrections.

I would recommend more English and style issues to be improved by a native English speaker at a final reading.

Overall, I congratulate the authors for preparing this article, which required a lot of work.

Author Response

Article "Development and multicenter validation of a novel immune inflammation-based nomogram to predict survival in Western resectable gastroesophageal adenocarcinoma (GEA): the NOMOGAST” by Salati et al. is an interesting work on an important subject i.e., the GEA relapses (>50%) after curative-intent resection. The authors directed their scientific efforts to validate a nomogram for more accurate individualized risk estimation in resected GEA, a topic that is not yet elucidated and remains open.

Based on the analysis of the medical records of a training cohort (TC) and a validation cohort (VC) of patients undergoing radical surgery for c/uT2-T4 and/or node-positive GEA, the authors have collected the variables considered potentially interesting. Effects of prognostic factors were evaluated, and a nomogram was drawn using the R software's package Regression Modeling Strategies (ver. 5.0-1), and the performance of the prognostic model was evaluated and validated by Salati et al.

Initial data of the two cohorts as well as the results of the modelling are presented and discussed in two tables and 4 figures. The final conclusions are presented. The article is not loaded with unnecessary information.

However, as minor problems, I would suggest the following corrections:

  1. To more explicitly introduce any abbreviations used right from the start, and a final table of Abbreviations is very welcome.

This has been added to the text.

  1. References should be presented throughout the article more accurately, enclosed in brackets and followed by a full stop.

Done.

So YES, that's right    .....     [1].

                         .........         [2-3].

Please, correct lines: 77, 79, 85, 88, 94, 96, 98 and so on …, until the end of the article!!!

  1. Discussions should be improved by showing all model shortcomings, how results were compared, and the limitations imposed by statistics and the model.

Done.

  1. Final conclusions should be better punctuated, to present the real scientific value of this model more accurately.

Done.

  1. Overall, the study is interesting and deserves to be published with these corrections.

I would recommend more English and style issues to be improved by a native English speaker at a final reading.

A general English language editing has been performed.

Overall, I congratulate the authors for preparing this article, which required a lot of work.